# Consensus Ensemble Multitarget Neural Network Model of Anxiolytic Activity of Chemical Compounds and Its Use for Multitarget Pharmacophore Design

**DOI:** 10.3390/ph16050731

**Published:** 2023-05-11

**Authors:** Pavel M. Vassiliev, Dmitriy V. Maltsev, Alexander A. Spasov, Maxim A. Perfilev, Maria O. Skripka, Andrey N. Kochetkov

**Affiliations:** 1Laboratory for Information Technology in Pharmacology and Computer Modeling of Drugs, Research Center for Innovative Medicines, Volgograd State Medical University, 39 Novorossiyskaya Street, Volgograd 400087, Russia; 2Department of Pharmacology and Bioinformatics, Volgograd State Medical University, 20 KIM Street, Volgograd 400001, Russia; 3Laboratory of Experimental Pharmacology, Volgograd Medical Research Center, 1 Pavshikh Bortsov Square, Volgograd 400131, Russia

**Keywords:** artificial neural networks, multitarget docking, consensus ensemble model, anxiolytic activity, multitarget pharmacophores

## Abstract

A classification consensus ensemble multitarget neural network model of the dependence of the anxiolytic activity of chemical compounds on the energy of their docking in 17 biotargets was developed. The training set included compounds thathadalready been tested for anxiolytic activity and were structurally similar to the 15 studied nitrogen-containing heterocyclic chemotypes. Seventeen biotargets relevant to anxiolytic activity were selected, taking into account the possible effect on them of the derivatives of these chemotypes. The generated model consistedof three ensembles of artificial neural networks for predicting three levels of anxiolytic activity, with sevenneural networks in each ensemble. A sensitive analysis of neurons in an ensemble of neural networks for a high level of activity made it possible to identify four biotargets ADRA1B, ADRA2A, AGTR1, and NMDA-Glut, which were the most significant for the manifestation of the anxiolytic effect. For these four key biotargets for 2,3,4,5-tetrahydro-11H-[1,3]diazepino[1,2-a]benzimidazole and [1,2,4]triazolo[3,4-a][2,3]benzodiazepine derivatives, eight monotarget pharmacophores of high anxiolytic activity were built. Superposition of monotarget pharmacophores built two multitarget pharmacophores of high anxiolytic activity, reflecting the universal features of interaction 2,3,4,5-tetrahydro-11H-[1,3]diazepino[1,2-a]benzimidazole and [1,2,4]triazolo[3,4-a][2,3]benzodiazepine derivatives with the most significant biotargets ADRA1B, ADRA2A, AGTR1, and NMDA-Glut.

## 1. Introduction

According to WHO data for 2020 [1], 4.4% of the world’s population suffers from depression and 3.6% from anxiety disorders, which is currently about 635 million people. Anxiety can be an appropriate response to stressful situations but is considered a pathologic disorder when it is disabling and difficult to control [2]. Anxiety disorders (generalized anxiety disorder, panic disorder/agoraphobia, social anxiety disorder, and others) are the most prevalent psychiatric disorders, and are associated with a high burden of illness [3]. The current therapy used for the treatment has many disadvantages, such as higher cost and severe adverse reactions, and has suboptimal efficiency [4]; thus, the search for novel approaches in therapy and more effective drugs still goes on.

Pharmacophore modeling is a modern in silico approach in drug discovery [5], which, due to its versatility, is widely used to study ligand–biotarget interactions, virtual screening of biologically active compounds, and direct design of molecules with high pharmacological activity. Structure-based modeling using protein–ligand interactions and ligand-based modeling using general chemical features are the two main approaches to building pharmacophores.

According to IUPAC, a pharmacophore is a purely abstract concept that allows one to describe the general possibilities of intermolecular interactions for a group of compounds with respect to their target [6]. From this definition, in particular, it follows that it is not always possible, especially when constructing multitarget pharmacophores, to represent them in the form of sufficiently accurate 3D visualized models. The problem of the accuracy of constructing and visualizing pharmacophores becomes very difficult to resolve if multitarget pharmacophores are constructed for several structurally and functionally different target proteins. Therefore, when constructing general pharmacophores, they are most often limited to two similar biotargets.

The review [7] considers methods of multitarget drug design based on pharmacophores and docking. It mainly describes the sequential or parallel use of monotargeted pharmacophores in virtual screening. A technique for constructing general pharmacophores is also presented, based on the assumption that the number of bound ligand conformations is limited, even for different targets. However, its successful application is shown only by the example of constructing a bivalent pharmacophore to search for compounds with dual activity against two proteins with similar functions: enzymes leukotriene A-4 hydrolaseLTA4H and non-pancreatic secretory phospholipase A_2_ NPS-PLA2. Pharmacophore is easily formed and visually determined, it contains two hydrophobic sites and a metal coordination center.

An increase in the accuracy of building visualized multitarget 3D pharmacophores is provided by the ELIXIR-A plugin for VMD, which is used to refine pharmacophore interaction points between several ligands and several receptors [8].

Currently, various machine learning methods are widely used in the pharmacological search for novel highly active compounds, including those based on artificial neural networktechnologies [9].

Interesting in this regard is the work [10], in which, using a deep learning neural network, a model was developed for generating libraries of compounds with a possible multitarget action based on several monotarget pharmacophores.

However, in the available literary sources, we were unable to find publications devoted to the construction of an ensemble multitarget neural network model that reflects the dependence of the anxiolytic activity of chemical compounds on the energy of their docking into a variety of relevant biotargets.

There are also no works devoted to the construction of multitargetpharmacophores of high anxiolytic activity based on such neural network models.

All of this determined the main goal of this study, which aimed at building a classification consensus ensemble multitarget neural network model of the dependence of the anxiolytic activity of chemical compounds on the energy of their docking into relevant biotargets. The aim of the study is also to identify, based on this model, multitarget pharmacophores of high anxiolytic activity, reflecting the universal features of the interaction of some derivatives of nitrogen-containing heterocyclic compounds with a set of the most significant relevant biotargets, taking into account the mutual action of biotargets against each other.

## 2. Results

### 2.1. Data Preparation

The relevance of 2057 targets associated with anxiety disorder was rated in relation to 15 studied chemotypes of nitrogen-containing condensed hetero-cyclic compounds, and the 14 most significant target proteins were selected. Thus, 14 valid 3D models of these 14 target proteins relevant to the anxiolytic activity of the 15 studied chemotypes were found (see Section 4.1 in Materials and Methods). It is for these 14 biotargets that there are experimental data on their significance for the formation of the anxiolytic activity of the studied compounds, and it is these biotargets that they act on.

A verification database was formed on the structure and gradation level of anxiolytic activity of 216 known compounds structurally similar to the 15 chemotypes under study. The database contains 30 compounds with high activity, 64 compounds with moderate activity, 67 compounds with low activity, 34 active compounds without indicating the level of activity, and 21 inactive compounds (see Section 4.2 in Materials and Methods).

The construction and optimization of 3D models of 216 known anxiolytic compounds from the generated database werecarried out (see Section 4.3 in Materials and Methods).

The key binding amino acids of 17 sites of 14 relevant target proteins were identified and 17 docking regions of small molecule ligands at the sites of 14 relevant target proteins were formed using 14 valid 3D models (see Section 4.3 in Materials and Methods).

Ensemble docking of optimized 3D structures of 216 known anxiolytic compounds (from the created database) structurally similar to the 15 studied chemotypes was performed in 17 sites of 14 valid 3D models of 14 relevant target proteins. An affinity matrix of these 216 compounds for 17 sites of 14 relevant biotargets was obtained, containing 3672 minimum docking energies ∆E (see Section 4.3 in Materials and Methods).

Based on the results of data preparation, a training set for neural network modeling was formed, including three indicators of the level of anxiolytic activity of 216 known compounds, 17 indicators of their affinity ∆E for 17 sites of 14 target proteins, and 7 sampling variables (see Section 4.4 in Materials and Methods).

### 2.2. Neural Network Modeling

In the process of training neural networks for each of the three activity levels and each of the seven sampling options, one neural network with the best accuracy was selected—a total of 21 neural networks (see Section 4.4 in Materials and Methods).

As a result, a consensus ensemble multitarget neural network model of the anxiolytic activity of chemical compounds structurally relevant to the 15 studied nitrogen-containing heterocyclic chemotypes was formed. The model includes three ensembles of seven neural networks in each ensemble for activity levels h/nh, hm/nhm, and a/na, within which the final conclusion about the activity level of the predicted compound is developedby generalizing, on the basis of consensus, the spectrum of predictive estimates obtained using seven neural networks for each of the three activity levels.

The accuracy indicators of the classification neural networks included in the ensembles are shown in Table 1. In all cases, there is a fairly high accuracy of training, testing, and validation.

For each activity level for each selected neural network, an assessment was made on the general set of the overall prediction accuracy F_0_, prediction accuracy of active compounds F_a_ (sensitivity), inactive compounds prediction accuracy F_n_ (specificity), and prediction accuracy according to the ROC-analysis data.

The consensus estimate of the level of anxiolytic activity of a particular chemical structure using the resulting ensemble of neural networks was performed as follows. For a predicted compound for each activity level, using seven neural networks, the presence or absence of this activity level was calculated, resulting in a spectrum from three sets of alternative predictive estimates. The processing of this spectrum of primary estimates was carried out using a simple unweighted consensus, where the result of voting on seven outcomes was taken to be the one for which four or more matching ratings were observed.

The obtained values of prediction ability are also shown in Table 1. In all cases, a fairly high accuracy of the prediction is observed.

The minimum classification accuracy was obtained for Activity level High, neural network No 427. In this case, ROC = 83.8%, its significance according to the binomial test *p* = 1.27 × 10^−12^ and Matthews correlation coefficient MCC = 0.752, and its significance according to Fisher’s exact test *p* < 5 × 10^−5^. All other indicators from Table 1 are of higher significance.

At the same time, all consensus prediction characteristics of the three ensembles of neural networks for all three levels of activity exceed the corresponding average values.

The applicabilitydomain for the resulting model is determined by the range ΔE from −0.5 to −11.8 kCal/mol.

Thus, the obtained consensus ensemble multitarget neural network model of the anxiolytic activity of chemical compounds structurally relevant to the 15 studied heterocyclic chemotypes is characterized by a high prediction ability.

### 2.3. Consensus Analysis of the Sensitivity of Neurons and Determination of Biotargets That Are the Most Significant for the Formation of High Anxiolytic Activity of Compounds Relevant to the Chemotypes of the Studied Derivatives of Nitrogen-Containing Condensed Heterocycles

The results of the consensus sensitivity analysis of neurons of seven neural networks with the best accuracy obtained in seven sampling variants for the high activity level (see Section 4.5 in Materials and Methods) are shown in Table 2.

Out of 17 neurons, only 4 showed stable high sensitivity in all 7 variants of the analyzed neural networks.

Thus, in relation to the 15 studied nitrogen-containing heterocyclic chemotypes, the most significant for the formation of a high level of anxiolytic activity of their derivatives are the ADRA1B, ADRA2A, and AGTR1 receptors and the glutamate site of NMDA receptor.It is these four biotargets that have the highest affinity for the eight studied compounds, and it is these biotargets that make the most significant contribution to the formation of the anxiolytic activity of the studied compounds.

### 2.4. Construction of Multitarget Pharmacophores of High Anxiolytic Activity of the Studied Derivatives of Nitrogen-Containing Condensed Heterocycles

Monotarget pharmacophores were constructed for four synthesized and experimentally studied novel 11-substituted 2,3,4,5-tetrahydro-11H-[1,3]diazepino[1,2-a]benzimidazoles of the DAB series with high anxiolytic activity, regarding the most significant biotargets ADRA1B, ADRA2A, AGTR1, and NMDA-Glut (see Section 4.6 in Materials and Methods), which are shown in Figure 1.

For four synthesized and experimentally studied novel3,6-disubstituted [1,2,4]triazolo[3,4-a][2,3]benzodiazepines of the RD series with high anxiolytic activity, monotarget pharmacophores were also constructed for the most significant biotargets ADRA1B, ADRA2A, AGTR1, and NMDA-Glut, which are shown in Figure 2.

For more on the interaction of the eight ligands with the sites of the four significant biotargets, see Section 4.6 in Materials and Methods.

Superposition of monotarget pharmacophores for 11-substituted 2,3,4,5-tetrahydro-11H-[1,3]diazepino[1,2-a]benzimidazoles of the DAB series and 3,6-disubstituted [1,2,4]triazolo[3,4-a][2,3]benzodiazepinesof the RD series, two multitarget pharmacophores of high anxiolytic activity were obtained, reflecting the universal features of interactions of these derivatives with the most significant biotargets ADRA1B, ADRA2A, AGTR1, and NMDA-Glut, which are shown in Figure 3.

Thus, by solving the inverse problem, based on the analysis of the consensus ensemble multitarget neural network model of the anxiolytic activity of chemical compounds structurally relevant to 15 nitrogen-containing heterocyclic chemotypes, 4significant biotargets were identified and multitarget pharmacophores of high activity were constructed, characterizing the general features of the interactionof 2,3,4,5-tetrahydro-11H-[1,3]diazepino[1,2-a]benzimidazole and [1,2,4]triazolo[3,4-a][2,3]benzodiazepine derivatives with ADRA1B, ADRA2A, AGTR1, and NMDA-Glut receptors.

## 3. Discussion

During the present study, the following results were obtained for the first time, which are characterized by high scientific novelty.

A verification database was created on the structure and level of anxiolytic activity of 663 experimentally studied known chemical compounds;A data set was formed on the structure and level of anxiolytic activity of 216 known compounds structurally similar to 15 studied chemotypes of nitrogen-containing heterocycles;Seventeen biotargets were identified that determine anxiolytic activity and are relevant to 15 studied chemotypes of nitrogen-containing heterocycles;A consensus ensemble multitarget neural network model of the dependence of the anxiolytic activity of chemical compounds on their docking energies in 17 relevant biotargets was built;The high recognition and prediction ability of the obtained consensus model is shown, and on average, according to the ROCanalysis, it is 98.9%;A consensus sensitive analysis of neurons of the obtained model was performed and four key biotargets were identified ADRA1B, ADRA2A, AGTR1, and NMDA-Glut, which are the most significant for the formation of anxiolytic activity;For 2,3,4,5-tetrahydro-11H-[1,3]diazepino[1,2-a]benzimidazole and [1,2,4]triazolo[3,4-a][2,3]benzodiazepine derivatives with high anxiolytic activity, eight monotarget pharmacophores were built, which determine the affinity for each of the most significant biotargets ADRA1B, ADRA2A, AGTR1, and NMDA-Glut;For 2,3,4,5-tetrahydro-11H-[1,3]diazepino[1,2-a]benzimidazole and [1,2,4]triazolo[3,4-a][2,3]benzodiazepine derivatives, two multitarget pharmacophores were constructed to determine the overall affinity for the most important biotargets ADRA1B, ADRA2A, AGTR1, and NMDA-Glut, which together, taking into account their mutual influence on each other, providehigh anxiolytic activity.

In our opinion, the most conceptual of the results obtained are the following.

For the first time, multitarget pharmacophores were constructed that determine the affinity of nitrogen-containing heterocyclic compounds for several of the most important biotargets of anxiolytic activity, taking into account the mutual influence of these biotargets on each other.

It should be emphasized that the constructed multitarget pharmacophores reflect the mutual influence of the most significant key biotargets on each other. This is due to the fact that the identification of these biotargets was carried out using the technology of artificial neural networks, the main fundamental principle of which necessarily involves taking into account the interconnections of all neurons with each other.

It should also be specially noted that the identification of pharmacophores of anxiolytic activity was first carried out on the basis of solving the inverse problem of analyzing a pre-built QSAR model. At the same time, the consensus ensemble multitarget neural network model of the anxiolytic activity of chemical compounds structurally relevant to 15 nitrogen-containing heterocyclic chemotypes was also constructed for the first time.

For the first time, monotarget pharmacophores were constructed that determine the affinity of nitrogen-containing heterocyclic compounds to the most important biotargets of anxiolytic activity separately.

It is characteristic that none of the constructed pharmacophores includes the entire condensed heterocyclic system. The interaction is provided by various pharmacophore fragments of tricyclic systems, as well as pharmacophore fragments of substituents.

Thus, both monotarget and multitarget pharmacophore interaction of the analyzed nitrogen-containing heterocycles with key biotargets of anxiolytic activity is multicenter and is determined by different poses of the molecule in the sites of biotargets.

A formed consensus ensemble multitarget neural network model of the anxiolytic activity of chemical compounds, which isbuilt on its basis of monotarget and multitarget pharmacophores, isused in the directed search for new derivatives of nitrogen-containing condensed heterocyclic compounds with high anxiolytic activity.

## 4. Materials and Methods

The general scheme of the study is shown in Figure 4.

### 4.1. Biotargets Relevant to the Anxiolytic Activity of the Studied Derivatives of Nitrogen-Containing Condensed Heterocycles

At this stage of the present study, a list of biotargets most significant for the formation of the anxiolytic activity of the studied derivatives of nitrogen-containing condensed hetero-cycles was formed.

In the information system Open Targets [11], 2057 targets associated with anxiety disorder were found. Additionally, a list of 2697 *Homo sapiens* target proteins from the original QSAR-DB of Microcosm BioS 20.6.6 program of IT Microcosm system [12], for which there were experimental data on various types of target activity of chemical compounds, was used.

As a result of comparing these two lists, with specification of data from available literature sources, a list of 92 human biotargets relevant to anxiolytic activity and having reliable confirmation in the form of experimental data on the activity of compounds was formed. Appendix A lists these anxiolytic biotargets.

A comparative analysis of the structures of the studied derivatives of nitrogen-containing condensed heterocycles made it possible to identify 15 chemotypes, the main of which are diazepinobenzimidazoles, quinoxalines, benzoimidazopyrazines, mercaptobenzimidazoles, and benzodiazepines. Each chemotype was assigned a weight proportional to the number of studied derivatives of this chemotype. Appendix A contains names, structures, and weights of chemotypes.

Cumulatively for all 15 chemotypes and each of the 92 relevant anxiolytic biotargets, by structural similarity method using the original Microcosm BioS 20.6.6 program of IT Microcosm 7.2 system [12], 15,468 average estimates of the level of target activity were calculated, and the interval of changes wasfrom Ind = +5 (very high) to Ind = −5.0 (inactive). The found values served as a metric for the selection of biotargets that are most correlated with a high level of anxiolytic activity of derivatives of the analyzed chemotypes. The 14 most significant biotargets were selected with an overall average activity level index for all chemotypesIndGen ≥ 0 and with the maximum average activity level index for a particular chemotypeIndMax ≥ 1, taking into account the literature data on the high level of the considered type of activity of derivatives of a particular chemotype found in the experiment: ADRA1A, ADRA1B, ADRA2A, ADRA2B, AGTR1, GABRA1/GABRB2/GABRG2, HTR1A, HTR2A, HTR4, HTR7, MTNR1A, MTNR1B, GRIN1/GRIN2A/GRIN2B, and SLC18A2 (generally accepted abbreviations for proteins are indicated). It is for these 14 biotargets that there are experimental data on their significance for the formation of the anxiolytic activity of the studied compounds, and it is these biotargets that they act on. Appendix A lists these significant biotargets.

When selecting valid 3D models of the 14 anxiolytic biotargets significant for the 15 studied chemotypes, the most complete experimental 3D models were considered available in databases UniProt [13], PDBe [14], and RCSBPDB [15]. In addition, for proteins consisting of a single subunit, in the AlphaFold database [16] homology-based theoretical models were found. A total of 43 3D models were processed for 14 relevant anxiolytic biotargets: 29 experimental X-ray diffraction and 14 theoretical homology plots. Appendix A provides the list of types and numbers of 3D models for each significant biotargets.

The following parameters of X-ray diffraction analysis served as the validity criteria for the experimental 3D models of target proteins: (1) the length of the modeled amino acid sequence; (2) resolution; (3) the number of fragments.

The following basic characteristics served as criteria for the validity of theoretical 3D models: (1) the length of the simulated amino acid sequence; (2) the percentage of identical amino acids in the modeled protein and the template protein; (3) the statistical reliability of the model, according to the totality of the calculated criteria for assessing the quality.

As a result, for 14 anxiolytic target proteins of *Homo sapiens* relevant to 15 studied chemotypes, 14 valid 3D models were found, a list of which is given in Appendix A.

### 4.2. A Verification Database on the Structure and Anxiolytic Activity of Known Compounds Relevant to the Chemotypes of the Studied Derivatives of Nitrogen-Containing Condensed Heterocycles

At this stage of the present study, a verification database was created, which included verified formulas of known compounds that are structurally similar to the studied chemotypes, indicating the verified ordinal values of anxiolytic activity obtained as a result of cluster analysis.

Information on the chemical structure and anxiolytic activity of known compounds was obtained from the ChEMBL database [17]. Chemical formulas from primary datasets were validated by chemists: bonds and valences were checked, polyatomic functional groups were expanded, and structures were brought to standard form. In accordance with drug-likeness rules, structures with a molecular weight of more than 1000 Da were excluded from consideration. All verified data were combined into one set.

Data on the anxiolytic activity of the found known compounds were obtained in 97 different methods for 5 different organisms. The entire array of biological information was checked for correctness. If uninterpretable values of anxiolytic activity were indicated for some structure, it was excluded from consideration.

Each record from the final data set contained the structural formula of the compound, the value of its anxiolytic activity, and a link to the literary publication in which these data were given. As a result, a verified array of information was created, which included the structures of 663 known compounds tested by the world scientific community for anxiolytic activity.

An expert analysis of data on the anxiolytic activity of 663 known compounds from the primary database obtained in 97 different methods was carried out. In terms of experimental testing, similar data were combined into 33 general groups, from 11 to 161 values in each group, in order to unify them.

In each such group, using the Statistica 7 program [18] using the k-means method, a cluster analysis of the quantitative values of anxiolytic activity was performed, divided into 5 groups of activity: high, moderate, low, active, inactive.

The resulting ordinal activity level labels were entered into the primary database and checked for noncontradiction. For structures that had several entries in the database with several gradations of activity, the average score was taken as the final level of anxiolytic activity.

As a result, a verification database was created containing verified structural formulas of 663 known compounds and unified ordinal estimates of their anxiolytic activity. A certificate of state registration was received for the specified database.

Using MC SimScaf 20.9.1 program of IT Microcosm 7.2 system [12], taking into account the weights of chemotypes, 216 known anxiolytic compounds were selected that were structurally similar to the 15 studied chemotypes. To obtain a representative set, it was assumed that the number of selected structures should be greater than the product of the number of chemotypes and the number of biotargets relevant to them (15 × 14 = 210).

The average QL-modified Tanimoto similarity coefficient [19] of the selected structures with each of the chemotypes ranged from 0.160 to 0.431. The program algorithm provided for the selection of only compounds with individual structural features, excluding the appearance of duplicate structures.

As a result, a verification database was created on the structure and anxiolytic activity of 216 known compounds structurally similar to 15 studied chemotypes, containing verified structural formulas and unified ordinal estimates of the anxiolytic activity of these substances. File Anxio216.sdf in Appendix A contains names, structures, and activity levels of these compounds.

### 4.3. Ensemble Docking of Known Compounds Tested for Anxiolytic Activity, Relevant to the Chemotypes of the Studied Derivatives of Nitrogen-Containing Condensed Heterocycles

At this stage of the present study, we calculated the affinity matrix of 216 known anxiolytic compounds, structurally similar to 15 studied chemotypes, to 17 sites of 14 human target proteins relevant to the anxiolytic activity of these chemotypes by docking the affinity matrix.

For all 216 known anxiolytic compounds structurally similar to the studied chemotypes, molecular mechanics methods using MarvinSketch 17.1.23 program [20], separately for each compound, 10 conformers with the lowest energy were built. The constructed conformers were optimized using the MOPAC2012 program [21] using the PM7 semiempirical quantum chemical method. Among 2160 optimized conformers, 1conformer with the lowest total energy was selected for each compound.

Three approaches were used to identify binding sites in targets: (1) literature data on amino acid point mutations in proteins; (2) using the LigPlot+ 1.4.5 program [22] to identify amino acids responsible for ligand binding in experimental 3D protein models; (3) literature data on molecular modeling of ligand binding to corresponding proteins. The simultaneous use of three approaches made it possible to ensure the reliability of determining the localization of sites and their key binding amino acids.

In three biotargets, two different binding sites were identified: for GABRA1/GABRB2/GABRG2 benzodiazepine-binding and GABA-binding; for HTR2A serotonin-binding and allosteric; for GRIN1/GRIN2A/GRIN2B Ca-channel blocking and glutamate-binding. Thus, for 14 biotargets, 17 binding sites of various types were identified.

Key amino acids of 17 binding sites of 14 human target proteins relevant to the anxiolytic activity of the studied chemotypes are given in Appendix A.

Ensemble docking of optimized 3D models of 216 known anxiolytic compounds structurally similar to 15 studied chemotypes into 17 binding sites of 14 valid 3D models of 14 human target proteins relevant to the anxiolytic activity of the studied chemotypes was carried out using PyRx 0.8 program [23] and AutoDock Vina 1.1.1 program [24], taking into account data on key amino acids of binding sites and 3D models of docking spaces covering the sites.

Separately, for each 3D model of each type of target protein and each 3D structure of 216 known anxiolytic compounds, docking was performed 5times, considering 10 energetically most favorable conformations of the ligand at the biotarget site. According to the obtained 50 values for the docking compound, the minimum energy of its docking ΔE for this type of biotarget was determined. In total, an array of 183,600 docking energies was processed in this way.

According to the docking data, an affinity matrix of 216 known anxiolytic compounds, structurally similar to 15 studied chemotypes, was formed in relation to 17 sites of 14 target proteins relevant to the anxiolytic activity of these chemotypes, including 3672 minimum values ∆E.

### 4.4. Neural Network Modeling of the Dependence of the Level of Anxiolytic Activity of Chemical Compounds on Their Affinity for Biotargets Relevant to the Chemotypes of the Studied Derivatives of Nitrogen-Containing Condensed Heterocycles

At this stage of the present study, a consensus ensemble multitarget neural network model of the anxiolytic activity of chemical compounds relevant to the studied chemotypes was built and the accuracy of the generated model was assessed. The training of neural networks was carried out using the Statistica 7 program [18] according to the scheme described in the work [25].

A meaningful, verified classification training set of 216 known compounds structurally similar to the 15 studied chemotypes, required for neural network modeling, in terms of the level of anxiolytic activity and affinity spectra, included (1) graded values of the anxiolytic activity of these compounds, obtained as a result of cluster analysis; (2) spectra of the minimum values of the energies of their interaction with 17 sites of 14 relevant biotargets, obtained as a result of ensemble docking; (3) sampling variables.

For use in the construction of neural network models, in addition to the level of anxiolytic activity “high” (“high”), the combined gradations of activity were introduced: “high or moderate” (“pronounced”) and “high or moderate or low or active” (“active”). For these gradations of activity, variables with index designations for the presence/absence of these activity levels were added to the primary table: h/nh, hm/nhm, and a/na, for the gradations “high”, “pronounced”, and “active”, respectively.

Seven sampling variables set the options for the formation of training, testing, and validation subsets in a ratio of 5:1:1 and were used to build different versions of neural networks. In the first such variable, five compounds in a row are assigned the value train, then one compound is assigned the value test, and one compound is assigned the value val; then the assignment block is repeated until all 216 compounds are exhausted. The second and subsequent sampling variables are formed as a result of a shift by one compound down the table.

The formed training set is a matrix of 216 rows with data on compounds, for each of which the following variables are defined in 29 columns:

Code—compound cipher number; Level—clustered activity level (high, moderate, low, inactive); LevH, LevHM, LevA—labels of combined activity gradations (h/nh, hm/nhm, a/na); ADRA1A, ADRA1B, …, SLC18A2—docking energy at 17 sites of target proteins, kCal/mol; Sample1, …, Sample7—sampling variables. The training set is presented in Appendix A.

In accordance with Kolmogorov’s theorem [26], a dependence of any complexity can be approximated using a two-layer artificial neural network. In this case, it was desirable to ensure the convolution of signals from a plurality of input neurons into a small number of intermediate patterns.

Therefore, a two-layer perceptron MLP k-m-2 with a bottleneck was chosen as the initial architecture of the neural network. Here, k is the number of input neurons, in this case, 17, according to the number of docking energies in 17 biotarget sites; m is the number of hidden neurons, set by the program from 3 to 15, since 2 < m < k. When constructing classification perceptron networks, it is most optimal to use Cross entropy as an error function [27]. In this case, the activation function for output neurons is a multivariate version of the Softmax logistic function, and for hidden neurons, the four most common activation functions Identity, Logistic, Tanh, Exponential are used.

Compared to other machine learning methods, a neural network with a multilayer perceptron architecture has the following advantages: (1) it takes into account the mutual influence of each of the input neurons (variables) on all other neurons; (2) it is insensitive to the presence of noise; (3) it processes input data of any type (binary, discrete, continuous); (4) it allows you to approximate any complex non-linear or discrete dependencies and build any separating functions [28,29,30].

Neural networks were trained for each pair of combined gradations h/nh, hm/nhm, a/na, using the back-propagation algorithm, by enumeration of four different activation functions for the hidden layer of neurons, using seven sampling options.

In order to achieve the best learning outcome, the number of networks for each of the three pairs of activity levels and each of the seven sampling options was set to 6000, with automatic selection in each training cycle of the 600 best networks. After the end of training for a given level of activity and a given sampling option, out of the 600 best neural networks selected by the program, according to the totality of characteristics of the accuracy of training, testing, and validation, 5 best were manually selected, of which 1best was manually selected in terms of prediction accuracy on the general set of a given activity level.

In total, about 125,000 neural networks were trained and 12,600 automatically selected neural networks were analyzed manually.

As a result, a consensus ensemble multitarget neural network model of the anxiolytic activity of chemical compounds structurally relevant to 15 studied chemotypes was formed. The model includes three ensembles of seven neural networks in each ensemble for activity levels h/nh, hm/nhm, a/na, in which the final conclusion about the activity level of the predicted compound is developedby generalizing, based on a two-level consensus, the spectrum of predictive estimates obtained using 21 neural networks for 3 levels of activity.

For each activity level for each selected neural network, indicators of the overall prediction accuracy were calculated on the training, test, and validation subsets, and an estimate was made on the general set of the overall prediction accuracy F_0_, the prediction accuracy of active compounds F_a_ (sensitivity), the prediction accuracy of inactive compounds F_n_ (specificity), and the prediction accuracy according to ROCanalysis.

Consensus estimate of the level of anxiolytic activity of a particular chemical structure using the resulting ensemble of neural networks was performed as follows.

For a predicted compound for each activity level, seven neural networks were used to calculate the presence or absence of this activity level, as a result of which a spectrum was obtained from three sets of alternative predictive estimates. The processing of this spectrum of primary estimates was carried out using a simple unweighted consensus, where the result of voting on seven outcomes was taken to be the one for which four or more coinciding estimates were observed.

Within the framework of this level 1 consensus model, for each gradation of activity for each ensemble of neural networks, an assessment was made on the general set of the overall accuracy of the consensus prediction F_0_, consensus prediction accuracy of active compounds F_a_ (sensitivity), consensus prediction accuracy of inactive compounds F_n_ (specificity), and the accuracy of the consensus prediction according to the ROCanalysis, which was performed using MedCalc 11.5.0.0 program [31].

The total consensus estimates of the 1st level should be noncontradictory [12]. A highly active compound corresponds to a set of consensus estimates of the 1st level “h hm a”; moderately active—a set of consensus estimates of the 1st level “nh hm a”; low active—a set of consensus estimates “nhnhm a”; inactive—a set of consensus estimates “nhnhmna”. Other sets of consensus estimates of the 1st level are contradictory; for such compounds, the prediction result was considered uncertain.

For the generated model, the applicability domain was calculated in accordance with «three sigma rule» [32] in terms of docking energies ΔE for all 17 sites of 14 relevant biotargets and all 216 compounds of the training set, as the minimum and maximum of all calculated boundary values.

### 4.5. Consensus Sensitive Analysis of Neurons and Determination of Biotargets Most Significant for the Formation of High Anxiolytic Activity of Compounds Relevant to the Chemotypes of the Studied Derivatives of Nitrogen-Containing Condensed Heterocycles

At this stage of the present study, a consensus sensitive analysis of neurons of the formed ensemble of neural networks was carried out and biotargets were identified that most significantly affect the formation of a high anxiolytic effect of the compounds relevant to the studied chemotypes. The sensitive analysis of 17 neurons of 7 neural networks built in seven samples for a high level of anxiolytic activity was carried out using the Statistica 7 program [18] according to the scheme described in [33]. The native values of the sensitivity indices of neurons of one network were converted into percentages relative to the sum of these indices.

When classifying a target as significant, the normalized value of 6% (100%/17, rounded) was taken as the limiting level of sensitivity of its neuron. Targets that were identified as key in at least five neural networks out of seven neural networks obtained in different samplings were considered to be consensus significant.

The consensus of sensitivity indicators calculated for several neural network models on different samples of the same training set allows for obtaining more accurate data on the significance of biotargets. At the same time, due to the main fundamental principle of all neural network technologies, the mutual influence of the found most significant neurons (biotargets) on each other is taken into account. It is these 4 found biotargets that have the highest affinity for the 8 studied compounds, and it is these biotargets that make the most significant contribution to the formation of the anxiolytic activity of these compounds.

### 4.6. Construction of Multitarget Pharmacophores of High Anxiolytic Activity of the Studied Derivatives of Nitrogen-Containing Condensed Heterocycles

At this stage of the present study, multitarget pharmacophores of high anxiolytic activity were constructed in relation to the most significant biotargets, taking into account their mutual influence on each other, based on information about the structure of newly tested derivatives of nitrogen-containing condensed heterocycles.

For two groups of synthesized andexperimentally studied novel highly active substances with laboratory codes, DAB 11-substituted 2,3,4,5-tetrahydro-11H-[1,3]diazepino[1,2-a]benzimidazoles [34,35] and RD3,6-disubstituted [1,2,4]triazolo[3,4-a][2,3]benzodiazepines [36] (4 compounds in each group) using the methods described above in this article, optimized 3D models were built and ensemble docking was performed. Using the LigPlot+ v2.2 program [22], for the most energetically favorable conformation at the site of a specific target, the key points of interaction between the structure of the analyzed molecule and the amino acids of this protein were determined. Appendix A provides the structures of highlyactive compounds with the key points of interaction in relation to the most significant biotargets. A total of 32 schemes reflecting the interaction of each of the eight new compounds with the site of each of the 4significant biotargets are given in Appendix A.

Separately, in each of the two groups of DAB and RD compounds, for each significant biotarget, monotarget pharmacophores of high anxiolytic activity were constructed by identifying common key points of interaction with a specific target for the compounds of this group. An interaction point was considered pharmacophore if it was determined to be binding in at least three out of four structures. Additionally, using the BIOVIA Discovery Studio 2021 program [37], for all found monotarget pharmacophores, their 3D representations were constructed, which were obtained as a result of the superposition of the most energetically favorable conformations (docking poses) of the studied compounds in the binding sites of the four most significant biotargets ADRA1B, ADRA2A, AGTR1, NMDA.

Two multitarget pharmacophores of high anxiolytic activity for a set of four significant biotargets in two groups of nitrogen-containing condensed heterocyclic compounds were constructed in a similar way, by generalizing the constructed monotarget pharmacophores. Using the BIOVIA Discovery Studio 2021 program [37] for these two found multitarget pharmacophores, their 3D representations were built, which were obtained as a result of the superposition of 3D representations of monotarget pharmacophores.

## 5. Conclusions

The final results of the study allow us to draw the following conclusions.

Seventeen relevant biotargets were identified that determine the anxiolytic activity of the studied 2,3,4,5-tetrahydro-11H-[1,3]diazepino[1,2-a]benzimidazole and [1,2,4]triazolo[3,4-a][2,3]benzodiazepine derivatives;Using the method of artificial neural networks, a consensus ensemble multitarget model of the dependence of the anxiolytic activity of chemical compounds on their docking energies in 17 relevant biotargets was built, the accuracy of which was 98.9%;Based on the consensus analysis of the sensitivity of neurons in the obtained model, four key biotargets ADRA1B, ADRA2A, AGTR1, and NMDA-Glut were identified, which are most significant for the formation of the anxiolytic activity of the studied compounds;For these biotargets ADRA1B, ADRA2A, AGTR1, and NMDA-Glut, eight monotargeted pharmacophores of the studied 2,3,4,5-tetrahydro-11H-[1,3]diazepino[1,2-a]benzimidazole and [1,2,4]triazolo[3,4-a][2,3]benzodiazepine derivatives were constructed;For the studied 2,3,4,5-tetrahydro-11H-[1,3]diazepino[1,2-a]benzimidazole and [1,2,4]triazolo[3,4-a][2,3]benzodiazepine derivatives by superposition of monotarget pharmacophores, two multitarget pharmacophores were constructed that determine the overall affinity for biotargets ADRA1B, ADRA2A, AGTR1, and NMDA-Glut in combination, taking into account their mutual influence on each other, which provide high anxiolytic activity;The formed consensus ensemble multitarget neural network model of anxiolytic activity of chemical compounds and the pharmacophores built on its basis are used in the directed search for new 2,3,4,5-tetrahydro-11H-[1,3]diazepino[1,2-a]benzimidazole and [1,2,4]triazolo[3,4-a][2,3]benzodiazepine derivatives with high anxiolytic activity.

## 6. Patents

Vassiliev, P.M.; Maltsev, D.V.; Perfilev, M.A.; Spasov, A.A.; Skripka, M.O.; Kochetkov, A.N. Compounds with anxiolytic activity. Certificate of state registration of the database RU2022621744 (Russia). Official Bulletin of Federal Institute of Industrial Property “Computer Programs. Database. Topologies of integrated microchips”. 2022, 7. https://www1.fips.ru/ofpstorage/Doc/PrEVM/RUNWDB/000/002/022/621/744/2022621744-00001/DOCUMENT.PDF (accessed on 25 August 2022).

## Figures and Tables

**Figure 1 pharmaceuticals-16-00731-f001:**
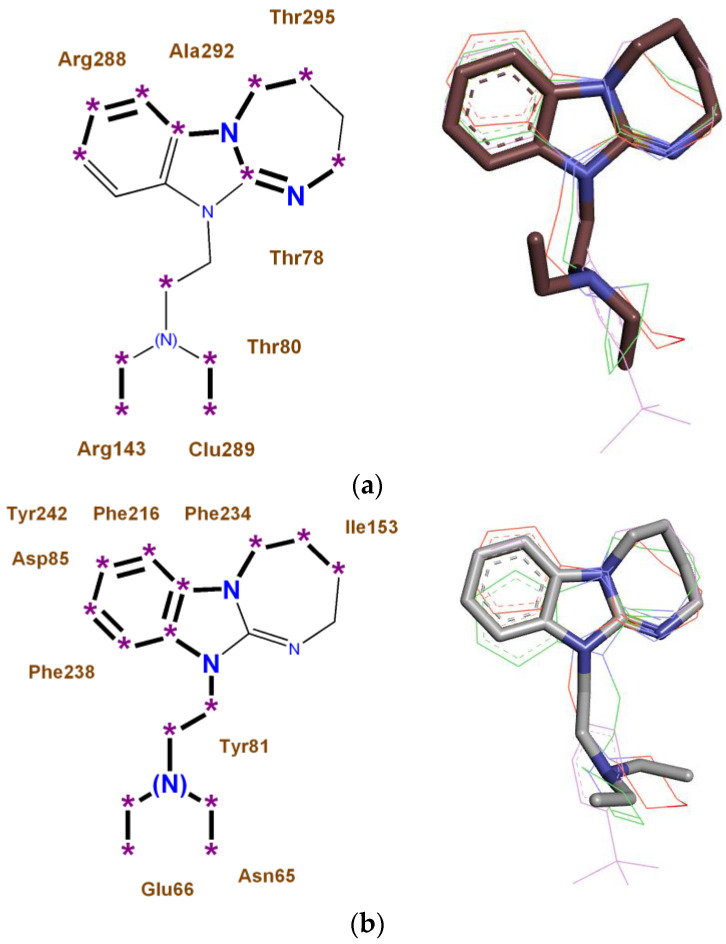
Monotarget pharmacophores of high anxiolytic activity of 11-substituted 2,3,4,5-tetrahydro-11H-[1,3]diazepino[1,2-a]benzimidazoles: (**a**) in relation to the ADRA1B receptor; (**b**) in relation to the ADRA2A receptor; (**c**) in relation to the AGTR1 receptor; (**d**) in relation to the NMDA receptor, glutamate site. *: Key binding points draw as stars. Key binding fragments draw in bold. Three-dimensionalrepresentations of pharmacophores are shown on the right.

**Figure 2 pharmaceuticals-16-00731-f002:**
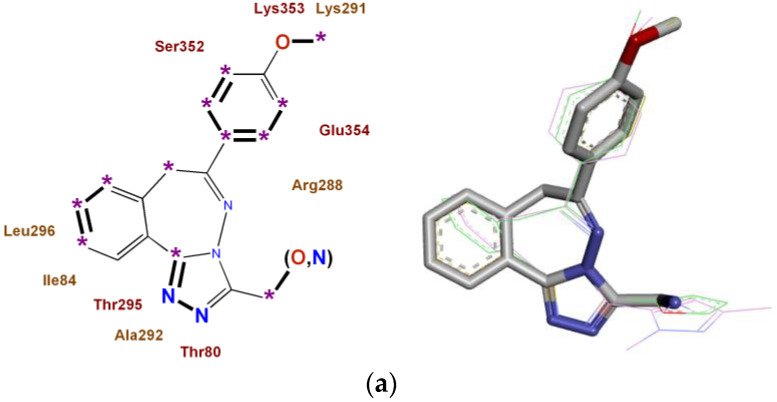
Monotarget pharmacophores of high anxiolytic activity of 3,6-disubstituted [1,2,4]triazolo[3,4-a][2,3]benzodiazepines: (**a**) in relation to the ADRA1B receptor; (**b**) in relation to the ADRA2A receptor; (**c**) in relation to the AGTR1 receptor; (**d**) in relation to the NMDA receptor, glutamate site. *: Key binding points draw as stars. Key binding fragments draw in bold.Three-dimensional representations of pharmacophores are shown on the right.

**Figure 3 pharmaceuticals-16-00731-f003:**
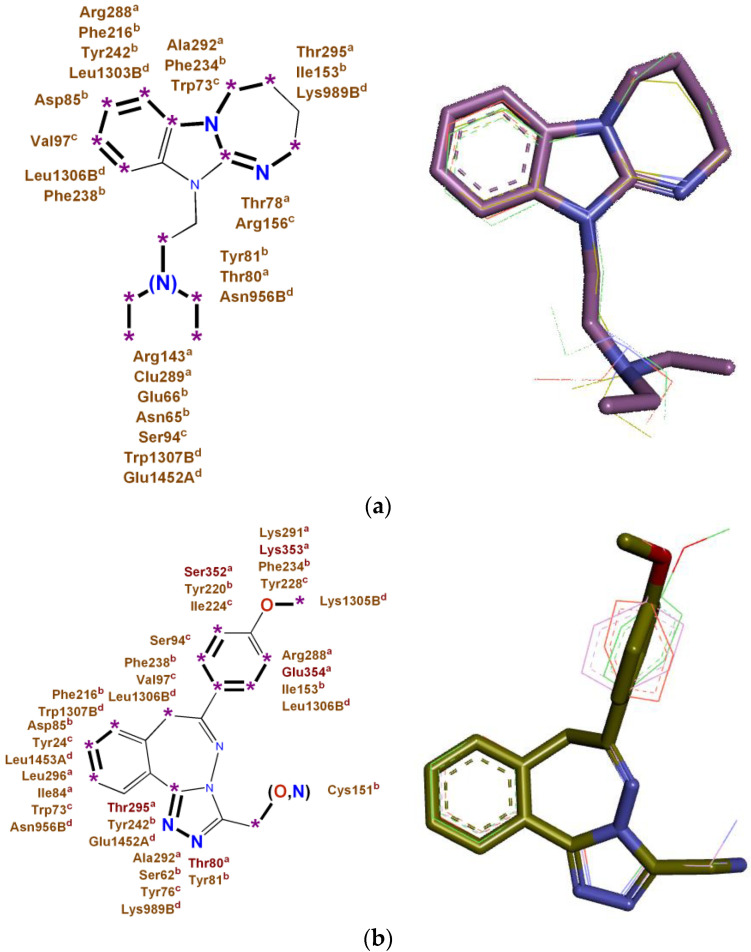
Multitarget pharmacophores of high anxiolytic activity, relevant in total to ADRA1B, ADRA2A, AGTR1, and NMDA-Glut biotargets: (**a**) for 11-substituted 2,3,4,5-tetrahydro-11H-[1,3]diazepino[1,2-a]benzimidazoles; (**b**) for 3,6-disubstituted [1,2,4]triazolo[3,4-a][2,3]benzodiazepines. Key binding fragments draw in bold. Key binding amino acids are indicated by superscripts: “a” for the ADRA1B receptor; “b” for the ADRA2A receptor; “c” for the AGTR1 receptor; “d” for the NMDA receptor, glutamate site. *: Key binding points draw as stars. Key binding fragments draw in bold. Three-dimensional representations of pharmacophores are shown on the right.

**Figure 4 pharmaceuticals-16-00731-f004:**
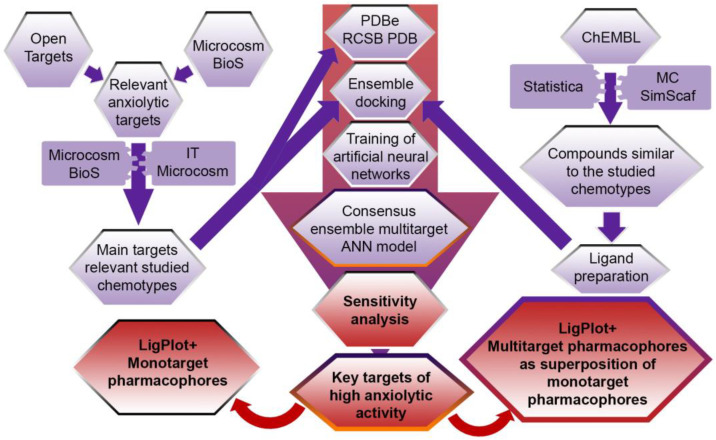
The general scheme for constructing multitarget pharmacophores of high anxiolytic activity based on consensus ensemble multitarget neural network model.

**Table 1 pharmaceuticals-16-00731-t001:** Accuracy indicators of the best neural network ensembles reflecting the multitarget dependence of the anxiolytic activity of chemical compounds structurally similar to 15 studied heterocyclic chemotypes on their docking energies in 17 sites of 14 relevant biotargets.

No ^1^	Architecture ^2^	Train ^3^	Test ^4^	Val ^5^	F_0_ ^6^	F_a_ ^7^	F_n_ ^8^	ROC ^9^
Activity level—High								
634	MLP 17-16-2 Tanh Softmax	100.0	74.2	90.0	94.9	90.0	95.7	91.5
39	MLP 17-16-2 Tanh Softmax	100.0	96.7	71.0	95.4	86.7	96.8	90.7
417	MLP 17-16-2 Tanh Softmax	99.4	74.2	80.6	93.1	83.3	94.6	89.1
250	MLP 17-16-2 Tanh Softmax	100.0	87.1	80.6	95.4	83.3	97.3	87.9
467	MLP 17-11-2 Tanh Softmax	99.4	74.2	90.3	94.4	80.0	96.8	84.8
427	MLP 17-16-2 Tanh Softmax	100.0	80.6	77.4	94.0	80.0	96.2	83.8
215	MLP 17-12-2 Tanh Softmax	100.0	77.4	80.6	94.0	90.0	94.6	92.5
Mean		99.8	80.6	81.5	94.5	84.8	96.0	88.6
Consensus		—	—	—	100.0	100.0	100.0	100.0
Activity level—High or Moderate								
219	MLP 17-14-2 Tanh Softmax	94.8	83.9	73.3	90.3	91.5	89.3	91.7
119	MLP 17-12-2 ExponentialSoftmax	99.4	86.7	61.3	92.1	90.4	93.4	92.5
390	MLP 17-15-2 Logistic Softmax	98.7	80.6	51.6	89.4	88.3	90.2	91.1
8	MLP 17-15-2 ExponentialSoftmax	98.7	90.3	64.5	92.6	91.5	93.4	93.1
587	MLP 17-16-2 Tanh Softmax	100.0	77.4	67.7	92.1	89.4	94.3	92.3
259	MLP 17-11-2 Tanh Softmax	98.1	77.4	67.7	90.7	94.7	87.7	93.0
49	MLP 17-17-2 Tanh Softmax	96.8	90.3	71.0	92.1	90.4	93.4	92.3
Mean		98.1	83.8	65.3	91.3	90.9	91.7	92.3
Consensus		—	—	—	97.2	96.8	97.5	97.2
Activity level—Active								
67	MLP 17-15-2 Logistic Softmax	99.4	93.5	83.3	96.3	97.9	81.0	93.6
497	MLP 17-7-2 Tanh Softmax	99.4	80.0	90.3	95.4	95.9	90.5	95.6
44	MLP 17-6-2 Tanh Softmax	97.4	100.0	87.1	96.3	96.9	90.5	96.4
222	MLP 17-7-2 Tanh Softmax	99.4	93.5	87.1	96.8	98.5	81.0	97.0
47	MLP 17-12-2 Logistic Softmax	100.0	100.0	90.3	98.6	99.0	95.2	95.8
206	MLP 17-11-2 Tanh Softmax	100.0	77.4	90.3	95.4	96.9	81.0	89.2
222	MLP 17-7-2 Tanh Softmax	96.8	93.5	87.1	94.9	95.4	90.5	94.6
Mean		98.9	91.1	87.9	96.2	97.2	87.1	94.6
Consensus		—	—	—	99.1	99.0	100.0	99.5
General mean		98.9	85.2	78.2	94.0	91.0	91.6	91.8
General mean for consensus		—	—	—	98.8	98.6	99.2	98.9

^1^ Number of the best neural network in training samplings 1…7. ^2^ Multilayer perceptron, the number of input–hidden–output neurons, the activation functions for hidden and output layers. ^3^ Training accuracy, %. ^4^ Accuracy in independent testing, %. ^5^ Validation accuracy in cross-leave Monte-Carlo control, %. ^6^ Total prediction accuracy on the combined set, %. ^7^ Prediction accuracy of active compounds (sensitivity) on the combined set, %. ^8^ Prediction accuracy of inactive compounds (specificity) on the combined set, %. ^9^ Accuracy according to ROCanalysis on the combined set, %.

**Table 2 pharmaceuticals-16-00731-t002:** Consensus analysis data of the sensitivity of neurons of the best neural networks, reflecting the dependence of high anxiolytic activity of chemical compounds structurally similar to 15 studied heterocyclic chemotypes on their docking energies in 17 sites of 14 relevant biotargets.

Target ^1^	634 ^2^	39 ^2^	417 ^2^	250 ^2^	467 ^2^	427 ^2^	215 ^2^	Sign ^3^
ADRA1A	5.7	3.3	3.4	4.6	6.8	3.7	5.8	1
ADRA1B	6.4	8.8	6.9	7.0	4.0	7.7	8.0	6
ADRA2A	7.6	12.4	11.0	6.0	6.7	7.5	4.1	5
ADRA2B	3.0	3.7	4.3	4.5	11.6	4.3	6.0	1
AGTR1	6.7	6.5	7.6	6.1	4.8	7.7	11.2	6
GABA-A-GABA	4.3	3.1	4.2	6.1	4.2	3.8	3.0	1
GABA-A-Benz	6.1	5.5	3.3	4.2	5.0	6.7	7.0	3
HTR1A	6.0	4.1	3.6	5.5	3.9	4.3	2.6	0
HTR2A-Spec	4.7	4.7	5.2	5.8	8.1	5.6	7.0	2
HTR2A-Allo	6.5	14.6	12.3	4.2	5.9	2.6	6.4	4
HTR4	10.9	6.8	2.0	6.1	5.4	5.0	5.6	3
HTR7	3.7	4.1	5.8	8.0	10.7	6.8	6.6	4
MTNR1A	4.4	3.4	3.0	4.5	4.0	4.0	4.2	0
MTNR1B	4.8	5.0	4.4	10.5	4.7	3.6	3.2	1
NMDA-Glut	12.1	5.7	15.8	7.3	3.2	12.6	11.6	5
NMDA-Ca	3.5	5.0	4.0	5.3	6.4	10.9	3.7	2
SLC18A2	3.4	3.4	3.3	4.1	4.5	3.1	4.0	0

^1^ Generally accepted abbreviations for proteins, with site reference if necessary. ^2^ Number of the best neural network for high level of anxiolytic activity, according to Table 1. ^3^ Number of significant neurons with sensitivity > 6% (italicized).

## Data Availability

Data arecontained within the article and Appendix A.

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
