# Peer review of "Consensus Ensemble Multitarget Neural Network Model of Anxiolytic Activity of Chemical Compounds and Its Use for Multitarget Pharmacophore Design"

_pharmaceuticals, 2023, doi:10.3390/ph16050731_

Round 1

Reviewer 1 Report

The manuscript is underdeveloped. The authors did not explain the importance of the targets and how multitarget compounds work.

Also, showing some possible structure is not enough, authors should demonstrate the synthetic feasibility by providing possible synthetic routes with suitable references.

Here is one example of how the authors presented their multitarget drug design approach.

J. Chem. Inf. Model. 2017, 57, 3, 403–412

The authors did not discuss the ADME and toxicity profile of the proposed compounds.

I believe the conclusion is also missing.

Author Response

Dear Colleague,
The authors thank You for Your valuable comments and questions, the consideration of which helped to improve the quality of the manuscript proposed for publication.
Please see the attachment.
Always Yours,
Pavel Vassiliev 

Reviewer 2 Report

Pavel M. Vassiliev et al. are describing an multitarget NN model for predicting anxiolytic activities. The developed model was further used to facilitate the pharmacophore design. The following are specific questions and comments. 

1. In Introduction, line 38. “… which is currently only about 635 million people”. The word “only” may downplay the significance of depression and anxiety disorders.

2. Section 2.1 is the Data preparation. But sources of targets and compounds are not specified. What are references or information sources to support selected 2057 anxiety disorder related targets? What are sources for authors to collect active compounds? Are there any compounds designed and generated from authors’ laboratories? Etc. Usually this info should be provided at the first time they are mentioned. At least authors should state that details are specified in “Materials and Methods”.

3. Ensemble docking is a key step in this study. Docking poses to show receptor-ligand interactions between representative compounds and critical binding amino acids across 17 sites of 14 relevant target proteins are expected.

4. Accuracy was used as the indicator in the model training process. But when we evaluate the performance of a neural network, metrics including precision, recall, MCC, etc. should be considered comprehensively. These metrics can evaluate the performance from diverse aspects.

5. Multilayer perceptron model was selected for the model training. What is the rationale behind this selection? How does the MLP compared to other mainstream ML methods like RF, SVM, logistic regression etc.?

6. Figures 1-3 demonstrate multitarget pharmacophores. Docking poses are highly desired to better illustrate how these key pharmacophores contribute to the binding. 

Overall, the reviewer would suggest a major revision if the above-mentioned comments can be addressed appropriately. 

Author Response

(The authors gave the same response as above.)

Round 2

Reviewer 1 Report

The authors considered most of the comments and suggestions. I believe the manuscript is acceptable for publication.

Reviewer 2 Report

Authors have addressed reviewer's comments and questions.